# Suppression of Rice Osa-miR444.2 Improves the Resistance to Sheath Blight in Rice Mediating through the Phytohormone Pathway

**DOI:** 10.3390/ijms24043653

**Published:** 2023-02-11

**Authors:** Tao Feng, Zhao-Yang Zhang, Peng Gao, Zhi-Ming Feng, Shi-Min Zuo, Shou-Qiang Ouyang

**Affiliations:** 1College of Horticulture and Plant Protection, Yangzhou University, Yangzhou 225009, China; 2Key Laboratory of Plant Functional Genomics of the Ministry of Education/Jiangsu Key Laboratory of Crop Genomics and Molecular Breeding, Agricultural College, Yangzhou University, Yangzhou 225009, China

**Keywords:** rice, microRNA, sheath blight disease, resistance, plant hormones

## Abstract

MicroRNAs (miRNAs) are a class of conserved small RNA with a length of 21–24 nucleotides in eukaryotes, which are involved in development and defense responses against biotic and abiotic stresses. By RNA-seq, Osa-miR444b.2 was identified to be induced after *Rhizoctonia solani* (*R. solani*) infection. In order to clarify the function of Osa-miR444b.2 responding to *R. solani* infection in rice, transgenic lines over-expressing and knocking out Osa-miR444b.2 were generated in the background of susceptible cultivar Xu3 and resistant cultivar YSBR1, respectively. Over-expressing Osa-miR444b.2 resulted in compromised resistance to *R. solani*. In contrast, the knocking out Osa-miR444b.2 lines exhibited improved resistance to *R. solani*. Furthermore, knocking out Osa-miR444b.2 resulted in increased height, tillers, smaller panicle, and decreased 1000-grain weight and primary branches. However, the transgenic lines over-expressing Osa-miR444b.2 showed decreased primary branches and tillers, but increased panicle length. These results indicated that Osa-miR444b.2 was also involved in regulating the agronomic traits in rice. The RNA-seq assay revealed that Osa-miR444b.2 mainly regulated the resistance to rice sheath blight disease by affecting the expression of plant hormone signaling pathways-related genes such as ET and IAA, and transcription factors such as WRKYs and F-boxes. Together, our results suggest that Osa-miR444b.2 negatively mediated the resistance to *R. solani* in rice, which will contribute to the cultivation of sheath blight resistant varieties.

## 1. Introduction

Plants are constantly threatened by pathogens from the environment affecting both crop production and quality. Plants perceive the attack of pathogens by initiating a first line of innate immunity pattern triggered immunity (PTI) to recognize pathogen-associated molecular patterns (PAMPs) via pattern recognition receptors (PRRs). For example, FLAGELLIN SENSING 2 (FLS2) recognizes the bacterial flagellin peptide 22 in *Arabidopsis* [1]. Some plant cells secrete chitinase to digest chitin from pathogens, major constituents of the cell wall [2]. Pathogens, on the other hand, deliver effectors to counter the PTI. The secondary immunity is activated by the disease resistance (R) gene, leading to effector-triggered immunity (ETI) during this endless arms race. To cope with environmental stresses, plants have developed and modulated sophisticated mechanisms including plant hormones and transcription factors to complete their life cycle.

Rice (*Oryza sativa* L.) is one of the most important crops, and supports half of the world’s population. Rice production is often threatened by a variety of diseases including rice sheath blight caused by *Rhizoctonia solani Kühn* (*R. solani*), a soil-borne fungal pathogen. Rice sheath blight disease occurs throughout the rice growth stages and causes severe yield loss from 10 to 30%, and this increases to 50% under favorable conditions. In the fields, the pathogen initially infects the rice bottom sheath at the late tillering stage, and then mainly develops upward (slightly downward) along the sheath. So far, due to the lack of stable high resistance sources in rice germplasm, breeding for *R. solani* resistant rice cultivar progressed slowly [3]. There is a huge challenge in agriculture to control *R. solani* due to the broad host range and tenacious survivability in nature [3,4,5]. Although the application of fungicides is an effective strategy to control rice sheath blight, genetic engineering for resistance is the most environmentally and human friendly strategy to prevent the outbreak of the disease.

MicroRNAs (miRNAs) are a class of non-coding single stranded RNAs usually of 21–24 nucleotides in length that suppress target gene expression through post-transcriptional regulation and/or repressing translation via binding complementary sequences and accordingly participating in plant development and defense responses [6]. MiR393, the first discovered miRNA in *Arabidopsis*, was up-regulated by a flagellin-derived peptide, resulting in suppressing the expression of the auxin receptors TIR1, AFB2 and AFB3 upon *Pseudomonas syringer* infection of *Arabidopsis* [7]. In *Brassica* and *Arabidopsis*, miR1885 was verified as a functional miRNA induced specifically by Turnip mosaic virus (TuMV) infection and targeted TIR–NBS–LRR class disease-resistant transcripts [8]. Our previous data showed that SlymiR482e-3p mediated the resistance to *Fusarium oxysporum* by targeting NBS-LRR like gene *FRG3* in tomato [9]. Subsequently, studies over recent decades have reported that miRNAs are associated with rice resistance against pathogens such as the blast fungus *Magnaportheoryzae* (*M. oryzae*) [10,11] and rice stripe virus (RSV) [12]. Osa-miR398b was involved in rice immunity against *M. oryzae* by targeting multiple superoxide dismutase genes (*SOD*), including Cu/Zn-Superoxidase Dismutase1 (*CSD1*), *CSD2* and Os11g09780, (Superoxide Dismutase X, *SODX*), resulting in increased total SOD activity [13]. Moreover, over-expressing Osa-miR164a showed significant susceptibility to *M. oryzae* by targeting *OsNAC60*, which positively regulated innate immunity [14]. Similarly, over-expressing Osa-siR109944 enhanced resistance to *R. solani* in rice [15]. In addition, Osa-miR528 negatively regulated viral resistance in rice by cleaving L-ascorbate oxidase (AO) messenger RNA leading to the accumulation of reactive oxygen species. Upon viral infection, Osa-miR528 is preferentially associated with AGO18 to elevate AO activity and enhanced antiviral defense [12].

In our previous study, we found that several miRNAs had been regulated upon *R. solani* infection by sRNA-seq [16]. Here, by taking advantage of transgenic plants, our data showed that Osa-miR444b.2 played a critical role in response to the basidiomycete *R. solani* infection in rice plants. Our study indicated that Osa-miR444b.2 could be used as a target in crop breeding to improve resistance.

## 2. Results

### 2.1. Characterization of Osa-miR444b.2 in Response to R. solani Infection

To assess the response to pathogeninfection in rice, we used matchsticks colonized with the mycelium of *R. solani* isolate YN-7 to inoculate the stems of susceptible cultivar Xudao 3 (Xu3) and a resistant t cultivar, YSBR1. The sheaths of both cultivars presented yellowing and water soaked lesions at the first day post-inoculation (dpi), however, the necrosis lesion of Xu3 developed significantquicker than YSBR1 during the observed 25 days (Figure 1A,B). Our results indicated that YSBR1 possessed extraordinary resistance to *R. solani* compared to Xu3.

Our previous data showed that several miRNAs responded to the infection of *R. solani* in rice [16]. We further confirmed that the expression level of Osa-miR444b.2 was induced upon *R. solani* infection at the early stage (Figure 1C). In order to investigate the biological function of Osa-miR444b.2 in rice immunity, we constructed the transgenic lines over-expressing Osa-miR444b.2 (named as Osa-miR444b.2_OE) and knocking out Osa-miR444b.2 (named as Osa-miR444b.2_KO) in the Xu3 and YSBR1 genetic background, respectively (the primers used in this study are listed in Appendix A). The T1 generation of transgenic lines were obtained and confirmed by PCR and quantitative Real-time PCR (qRT-PCR). Sequencing the CRISPR-Cas9 editing-site of Osa-miR444b.2_KO lines results showed that nucleotide-deletion/insertions were presented in indifferent Osa-miR444b.2_KO lines (Figure 2A). The expression levels of Osa-miR444b.2 were further detected by qRT-PCR in both the KO and OE lines, respectively. The relevant transgenic lines were subsequently selected for further study (Figure 2B,C).

### 2.2. Overexpressing Osa-miR444b.2 Partially Compromises Sheath Blight Resistance in Rice

The invasion of *R. solani* initiates at the rice sheath during the late tillering stage, and the symptoms under field conditions appear with ellipsoid lesions, eventually causing dark borders as they mature. In the Xu3 genetic background, Osa-miR444b.2_KO transgenic lines presented slower ellipsoid lesion expansion than Xu3 control under the infection of *R. solani* (Figure 3). On the contrary, Osa-miR444b.2_OE transgenic lines were shown to be more sensitive to the infection of *R. solani* with large disease lesions and faster fungal expansion than the Xu3 control at 15 dpi (Figure 3). In the YSBR1 genetic background, over-expressing Osa-miR444b.2 resulted in a slight susceptibility to *R. solani* infection compared to YSBR1. However, the development of sheath blight symptoms was unchanged between Osa-miR444b.2-KO lines and YSBR1 (Appendix A). These results collectively indicated that Osa-miR444b.2 negatively manipulated the response of the sheath to *R. solani* infection in rice.

### 2.3. Osa-miR444b.2 Regulates the Agronomic Traits of Rice

The agronomic traits of the Osa-miR444b.2 transgenic lines, including plant height, tiller number, panicle length, 1000-grain weight and primary branch number were also investigated. In the Xu3 background, both the over-expressing and the knocking out of Osa-miR444b.2 had an impact on the agronomic traits. In particular, the Osa-miR444b.2_OE transgenic lines showed decreased plant height, tiller number, shortened panicle length, primary branch number, but increased 1000-grain weight. Contrarily, knocking out Osa-miR444b.2 resulted in increased plant height, tiller number, and 1000-grain weight (Figure 4 and Appendix A). To transgenic lines in the YSBR1 background, however, only the tillering was slightly repressed in Osa-miR444b.2-OE lines (Appendix A). With these data, we speculate that Osa-miR444b.2 may be involved in the development of rice but are likely cultivar dependent. Further study is needed to provide greater detail with regard to the potential molecular mechanisms.

### 2.4. Osa-miR444b.2 Mediates the Response to R. solani Infection by Affecting Plant Hormone Signal Transduction Pathways and Transcription Factors

To explore the potential mechanisms of how Osa-miR444b.2 manipulates the response to *R. solani* infection in rice, we generated RNA-seq libraries using Osa-miR444b.2 knockouts in the Xu3 background and over-expressings in YSBR1 background. Above libraries were named as Osa-miR444b.2_KO_0, 8 and 16, Osa-miR444b.2_OE_0, 8 and 16, presenting infected sheaths at 0 hpi, 8 hpi, and 16 hpi, respectively.

To KO transgenic lines, the Kyoto Encyclopedia of Genes and Genomes (KEGG) pathway enrichment analysis was conductedbased on the differentially expressed genes (DEGs) in the libraries Osa-miR444b.2_KO at 8 hpi and 16 hpi, respectively. Intriguingly, the DEGs in plant hormone signal transduction pathways, such as abscisic acid (ABA), ethylene (ET), indoleacetic acid (IAA) and jasmonic acid (JA), were dominant (Figure 5A,B). Furthermore, the DEGs of WRKY and F-box transcription factors (TFs) were also found to be changed, suggesting that these genes are likely to be involved in the response to *R. solani* infection (Figure 5C).

Similarly, the analysis of libraries from Osa-miR444b.2_OE was conducted. The KEGG pathway enrichment analysis showed that the DEGs in plant hormone signal transduction pathways including ABA, ET, IAA and JA were involved in the response to *R. solani* (Appendix A). Furthermore, the DEGs of transcription factors in miR444_OE were also shown to be involved in the response to *R. solani* infection (Appendix A).

Subsequently, we selected several DEGs forfurther confirmation. The qRT-PCR results showed that *Os01g32380*, a homologous gene of the NRR repressor, was suppressed gradually after *R. solani* infection. *Os01g72910* and *Os02g33820*, genes in the ABA signaling pathway, were induced at 8 hpi and then suppressed at 16 hpi. *Os03g18850*, a JA inducible pathogenesis-related class 10 gene, was induced at 8 hpiand thenat 16 hpi. *Os11g37950* and *Os12g36850*, disease related genes, were induced upon the infection of *R. solani*. *Os09g25070*, a WRKY transcription factor, was also significantly suppressed. *Os08g38990*, encoding the WRKY 30 protein, was suppressed continuously upon *R. solani* infection (Figure 6).

## 3. Discussion

MiRNAs not only play an important role in regulating plant growth and development, but also participate in plant defense processes and regulate immune responses to pathogen attack by manipulating target genes. Numerous studies have shown that miRNAs are involved in the immune response of rice to pathogens, such as *M. oryzae* [10,11,13,14], RSV [12], *Rice ragged stunt virus* (RRSV) [17] and *Xoo* [18]. However, the molecular mechanisms of miRNAs in response to *R. solani* infection in rice have been rarely reported so far. In our previous study, Osa-miR444b.2 was identified as respondingto the infection of *R. solani* at an early stage [16]. Here, Osa-miR444b.2-KO/OE transgenic materials have been constructed to explore Osa-miR444b.2 as a negative regulator in rice immunity against *R. solani*.

In rice, the miR444 family, which is specific to monocots, consists of six members (Osa-miR444a, b, c, d, e and f), and each Osa-miR444 precursor is processed into two or three sibling-miRs, producing a total of 12 mature miR444 isoforms [19]. Among them, Osa-miR444b and miR444a, only one base substitution, were reported as functioning in response to environmental stress. Rice miR444a plays multiple roles in the rice NO_3_-signaling pathway in nitrate-dependent root growth, nitrate accumulation and phosphate-starvation responses by targeting MADS-box genes [20,21]. Over-expressing Osa-miR444 results in enhanced resistance against RSV infection accompanied by the up-regulation of *OsRDR1* expression causing the RNA silencing of RSV [22]. In this study, however, our data showed that the expression of Osa-miR444b.2 was induced upon *R. solani* infection (Figure 1C), and over-expressing Osa-miR444b.2 resulted in enhanced susceptibility to *R. solani* infection (Figure 3A). These results indicated that over-expressing Osa-miR444b.2 weakened rice immunity leading to facilitate *R. solani* infection. Furthermore, Osa-miR444b.2 was reported as a negative factor in the mediation of rice immunity against *M. oryzae* by down-regulating the expression of MADS-box genes [23].

MiRNAs are also found to regulate the agronomic traits of rice. The over-expression of Osa-miR444a resulted in reduced tiller numbers in rice due to the repression of *OsD14* by the target gene *OsMADS57* [24]. Similarly, over-expressing Osa-miR1871 reduced panicle numbers by suppressing the expression of *OsLRK2* and *OsAAP5* (both are tiller-related genes) [10]. Moreover, rice Osa-miR393 and Osa-miR172 positively participated in flowering time and development [25]. Here, our data confirmedthat over-expressing Osa-miR444b.2 lead to suppressed tiller numbers as well as shorter panicle length, lower plant height, and lower primary branches. In contrast, it increased 1000-grain weight (Figure 4). Based on the above results, we concluded that Osa-miR444b.2 acted as a negative regulator, impacting agronomic traits.

Upon fungal pathogen attack, rice plants initiate a complicated but sophisticated defense response at molecular, biochemical and physiological levels. Phytohormones such as ABA, IAA/auxin, ET and JA have been shown to play conserved and divergent roles in fine-tuning immune responses in plant host [26]. The IAA functions as ‘molecular glue’ to mediate the interaction amongthe auxin receptors. Osa-siR109944 is specifically suppressed upon *R. solani* infection, and possesses a conserved function in interfering with plant immunity by affecting auxin homeostasis in rice plants [15]. In addition to regulating plant defense responses, auxin can also precipitate disease susceptibility against various pathogens [25,27]. ET plays an important role in rice basal immunity against the fungal pathogen invasion by positively or negatively modulating the host disease resistance [28]. Our collective transcriptome data demonstrated that Osa-miR444b.2 facilitated the invasion of *R. solani* by triggering rich diverse hormonal disorders in rice. The plant hormone signaling pathways were orchestrated by mediating the transcript level of some DEGs which is far exceed other detected pathways. (Figure 6A and Appendix A). Based on this, we speculated that an internal unification between how *R. solani* enhances host susceptibility by manipulation of hormone signaling and how rice attenuates this manipulation might be referred to as an arms race.

Furthermore, our transcriptome data indicated that transcription factors (TFs) were also involved in the response to *R. Solani* infection in rice (Figure 6C and Appendix A). TFs are important players in the response to biotic stresses such as fungal pathogen infection [29]. These findings imply that rice plants have evolved distinct strategies to defend against *R. solani*, which might be mediated by diverseTFs. Therefore, further elucidation ofthe molecular mechanisms underlying the co-evolutionary interaction between *R. solani* and the rice host will facilitate the development of novel strategies to manage sheath blight disease.

## 4. Materials and Methods

### 4.1. Plant Materials and Growth Conditions

Rice (Oryza sativa) susceptible cultivar Xu3 and resistant cultivar YSBR1 were employed to construct Osa-miR444b.2 over-expression and knockout of transgenic lines, respectively. Rice plants were cultivated under natural conditions to the early stage of booting, with normal fertilizer and water management during the period, and transplanted to a greenhouse (30 °C 14 h light (day)/24 °C 10 h darkness (night), 75–90% the relative humidity) at the tillering stage. Plants transplanted to the greenhouse were recovered for one week before inoculation of *R. solani*.

### 4.2. Pathogen Cultivation

The pathogen used in this study was the moderately pathogenic *R. solani* strain YN-7. The YN-7 strain was inoculated on a sterilized potato dextrose agar (PDA) medium, and cultured at 28 °C for 2–3 days in the dark. The vertical grain veneers were cut into chips with a length of 1 cm and a width of 0.3 cm, and immersed in potato dextrose broth (PDB) medium pre-inoculated with *R. solani*. The plates were incubated in the dark at 28 °C for 4 days until the hyphae covered the surface of the wood chips evenly. The wood chips were then ready for the later inoculation.

### 4.3. Pathogen Inoculation and Plants Material Collection

Rice plants at the late tillering stage were artificially inoculated with YN-7 using the method described previously [16]. Immediately after inoculation, all plants were placed in a plastic moist chamber with a humidity of 92–100% and temperature of (28 ± 2) °C. The sheaths were harvested at 5, 10 and 20 h post inoculation (hpi), respectively. In order to minimize experimental variations, all sheath samples consisted of five seedlings for each treatment. All experiments were repeated twice. The bottom sections of sheaths (2–3 cm) were frozen in liquid nitrogen and stored at −80 °C for total RNA extraction. Disease severity was rated by measuring lesion length at the indicated time points.

### 4.4. RNA Extraction and Northern Blot Analysis

Total RNA was extracted using the Trizol reagent (#15596026, Invitrogen, Carlsbad, CA, USA). Purified RNA was treated with DNase I (Thermo Fisher Scientific, Waltham, MA, USA). For each sample, all sheaths from three biological repeats were pooled together for total RNA extraction.

For northern blot analysis, 10 μg total RNA was resolved on urea denaturing polyacrylamide gels (Urea-PAGE). MiRNA-specific oligo nucleotide probes were end-labeled using γ32P-ATP (#M0201, New England Biolabs, Ipswich, MA, USA). U6 was used as the loading control. All blots were scanned by PhosphorImager (GE Life Sciences, Pittsburgh, PA, USA). The original digital images were cropped and adjusted in Photoshop CS6.

### 4.5. Gene Expression Assay

The genomic DNA of rice leaf sheath was extracted by the CTAB method [30] (Xia et al., 2012). DNA-free total RNA (1 μg) was subjected to RT using M-MLV (H-) Reverse Transcriptase (#R021-01, Vazyme, Nanjing, China) to produce cDNA products following the supplier’s protocol. A qRT-PCR reaction was performed using ChamQSYBR qPCR Master Mix (#Q311-02, Vazyme, China). Three replicates were performed for each gene. The rice 18S rRNA was used as the internal reference to normalize the relative expression level of the individual gene.

### 4.6. The CRISPR/Cas9 Plasmid Construct and Osa-miR444b.2 Transgenic Rice Acquisition

CRISPR (Clustered regularly interspaced short palindromic repeats)/Cas9 technology was used to construct the miRNA knockout vector [31]. Briefly, the selected specific18 bp sequence of the miRNA gene was added with the NGG base at the end. Homologous recombination was used to generate the pOs-sgRNA vector followed by generating the CRISPR/Cas9 vector of Osa-miR444b.2 using the Gateway LR reaction. Gene specific primers were designed to amplify the precursor sequence fragment of Osa-miR444b.2, and they were then homologously recombined into pCUbi1390 to obtain the over-expression vector of Osa-miR444b.2 in Xu3 and YSBR1, respectively.

### 4.7. RNA-Seq and Data Analysis

Total RNA from the leaf sheath infected with/without *R. solani* at 0, 8 and 16 hpi were used for the construction of transcriptome sequencing libraries named as miR444_KO_0, miR444_KO_8, miR444_KO_16, miR444_OE_0, miR444_OE_8, and miR444_OE_16, respectively. Primary sequencing data, known as raw reads, were produced by an Illumina Genome Hiseq4000, and were subjected to quality control (QC) to determine reliability. Subsequently, the filtered raw reads were aligned to reference sequences. All sequence reads were trimmed to remove the low-quality sequences. The sequence data were subsequently processed using the in-house software tool SeqQC V2.2. Using edgeR software, the ratio of gene expression levels in the treatment group and the control group was compared. The threshold for significantly differential expression was set as *p* value < 0.05 and fold change > 2. A GO enrichment and KEGG pathway enrichment analysis of DEGs were performed accordingly.

## 5. Conclusions

Taken together, the data reported in this studyindicated that Osa-miR444b.2 responded to the infection of *R. solani*, a basidiomycetous necrotroph causing rice blast disease. The expression of Osa-miR444b.2 was induced, and the over-expression of Osa-miR444b.2 impaired rice resistance against *R. solani* infection accompanied by the participation of plant hormones and TFs. Typically, our data also demonstrated that altering the expression level of Osa-miR444b.2 resulted in changing multiple agronomical traits. Our study provides a useful reference for rice breeding to enhance sheath blight resistance, and this module would broaden the understanding of immunity and growth in future studies focusing on the complicated interactive relationships between Osa-miR444b.2 and downstream signal transduction.

## Figures and Tables

**Figure 1 ijms-24-03653-f001:**
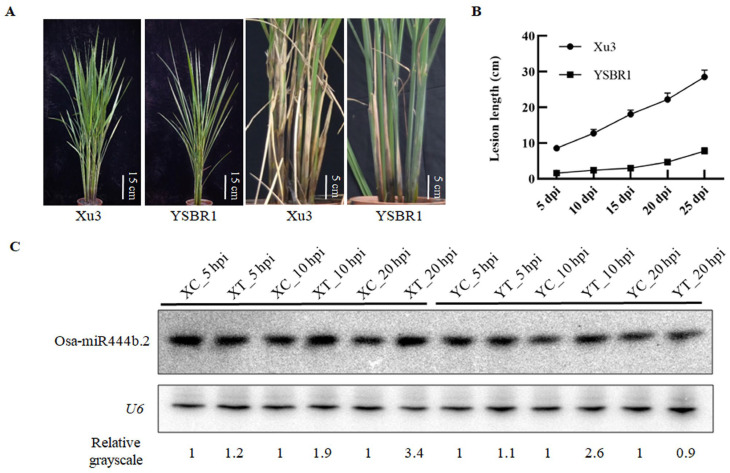
Rice sheath blight disease symptoms and Osa-miR444b.2 was induced by *R. solani* infection. (**A**) *R. solani* inoculation and disease severity of susceptible cultivar Xu3 and resistant cultivar YSBR1. (**B**) Statistics of lesion length in Xu3 and YSBR1 at different time points. Data are Mean ± SD (four independent plants). (**C**) The transcript level of Osa-miR444b.2 wasinduced by the infection of *R. solani*. Total RNA samples (10 μg) were used for northern blot analysis. The loading control was presented using a *U6* probe. Blots were imaged using a phosphorimager, and measured to the grey density using Image J software. The numbers below each blot presented the relative enrichment of Osa-miR444b.2 in each treatment normalized to the corresponding water-treated control. XC: Xu3 treated by water as control, XT: Xu3 infected by *R. Solani*, YC: YSBR1 treated by water as control, YT: YSBR1 infected by *R. Solani*.

**Figure 2 ijms-24-03653-f002:**
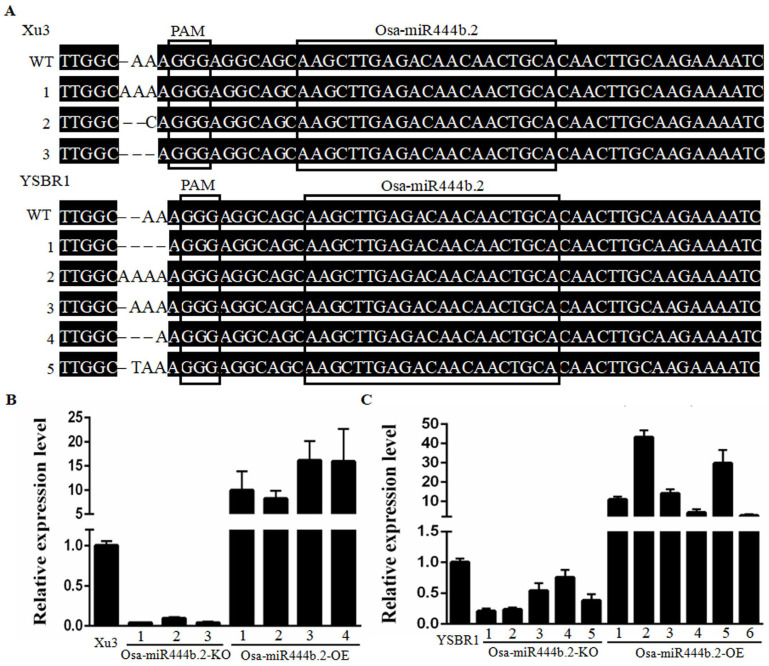
Constructions of Osa-miR444b.2 transgenic plants.(**A**) The sequencing results of Osa-miR444b.2 knocking out transgenic lines. The boxes represent the protospacer adjacent motifs (PAM) and the mature miRNA, respectively. (**B**) The relativeexpression levels of Osa-miR444b.2 in knocking out and over-expression materials in the Xu3 background. (**C**) The relative expression levels of Osa-miR444b.2 in knocking out and over-expression materials in the YSBR1 background.

**Figure 3 ijms-24-03653-f003:**
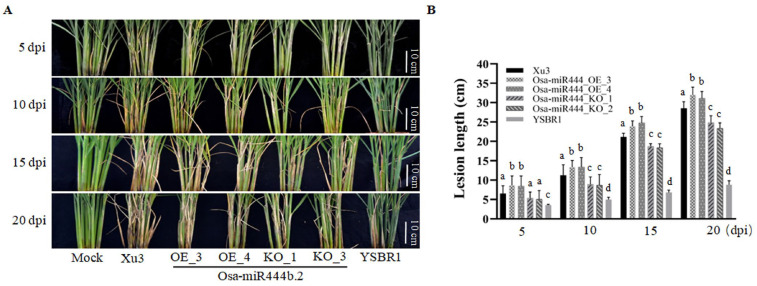
Osa-miR444b.2 enhances rice susceptibility to sheath blight in Xu3. (**A**) Sheath blight disease symptoms were photographed after the infection of *R. solani* at different time points. Scale bar, 10 cm. (**B**) The lesion lengths were scaled at different time points after the infection of *R. solani* at different time points. a presents the largest average, while bcd present significance (*p* < 0.01).

**Figure 4 ijms-24-03653-f004:**
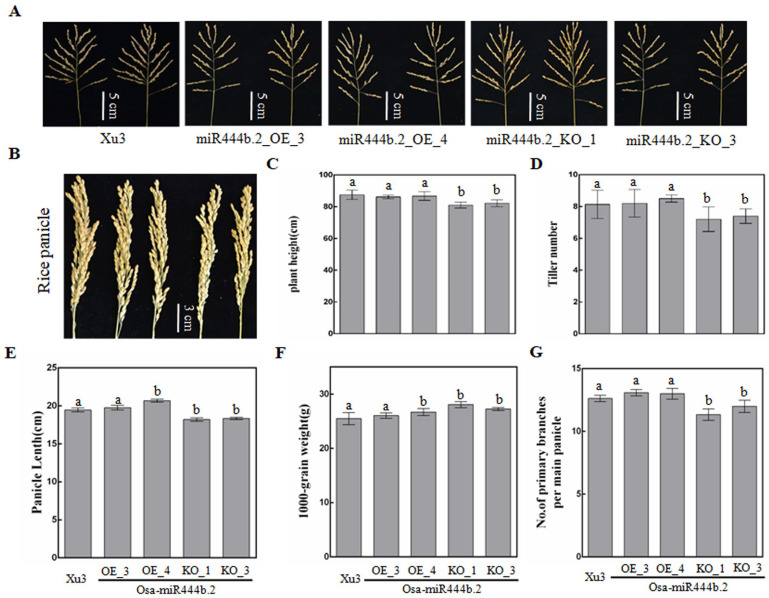
Effect of Osa-miR444b.2 expression level on agronomic traits in Xu3. (**A**) The morphology of primary branches of Osa-miR444b.2 knocking out and over-expression materials. (**B**) Panicle morphology of Osa-miR444b.2 knocking out and over-expression materials. (**C**) Statistics of the plant height. (**D**) Statistics of the tiller number. (**E**) Statistics of the panicle length. (**F**) Statistics of the 1000-grains weight. (**G**) Statistics of the number of primary branches panicle. a presents the largest average, while b presents the significant difference (*p* < 0.01).

**Figure 5 ijms-24-03653-f005:**
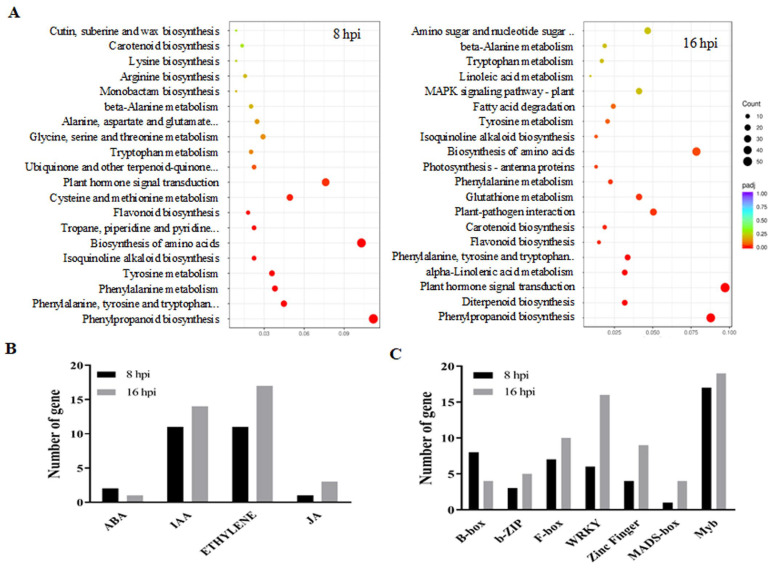
Osa-miR444b.2 participates in the response to *R. solani* infection by mediating the phytohormone signal transduction pathway and transcription factors. (**A**) Top 20 enriched KEGG pathways of differentially expressed genes in differential time. (**B**) The number of differentially expressed genes in plant hormone signal transduction. (**C**) The number of differentially expressed genes of TFs.

**Figure 6 ijms-24-03653-f006:**
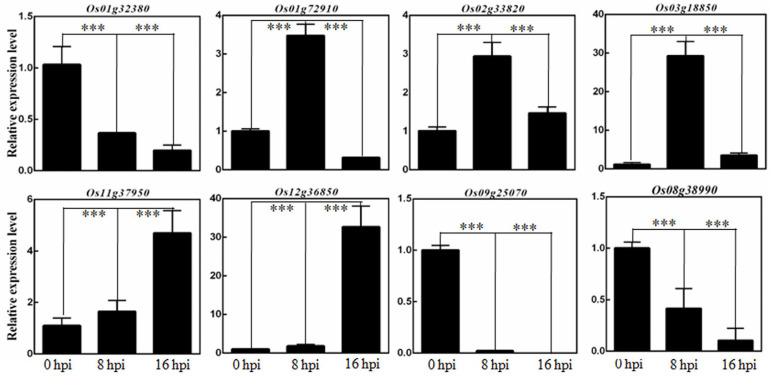
The relative expression levels of genes responding to *R. solani* infection. Annotation of selected genes, *Os01g32380*, encodingNRR repressor homologue 3 protein. *Os01g72910*, encoding ABA-stress-ripening-inducible 4 protein. *Os02g33820*, encoding Abscisic acid-stress-ripening-inducible 1 protein. *Os03g18850*, encoding Jasmonate inducible pathogenesis-related class 10 protein. *Os11g37950*, encoding pathogenesis-related gene 4Cprotein. *Os12g36850*, encoding pathogenesis-related gene 10B protein. *Os09g25070*, encoding WRKY 62protein. *Os08g38990*, encoding WRKY 30 protein. The significant difference analysis were performed. *** presents the significant difference (*p* < 0.01).

## Data Availability

The RNA-seq raw data in this study are available in the GenBank Nucleotide Sequence Databases with accession number PRJNA011642.

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
