# Peer review of "Suppression of Rice Osa-miR444.2 Improves the Resistance to Sheath Blight in Rice Mediating through the Phytohormone Pathway"

_ijms, 2023, doi:10.3390/ijms24043653_

Round 1

Reviewer 1 Report (New Reviewer)

The author found that Osa-miR444b.2 was induced by Rhizoctonia solani infection. Over-expressing Osa-miR444b.2 results in compromised resistance to R. solani, while knocking out Osa-21miR444b.2 improves the resistance to R. solani. These results indicate that Osa-miR444b.2 negatively regulates the resistance to R. solani in rice. In addition, Osa-miR444b.2 was also involved in regulating the agronomic traits. RNA-seq assay revealed that Osa-miR444b.2 may regulated the resistance to rice sheath blight disease by affecting the expression of plant hormone signaling pathways related genes, and transcription factors. I think this work is suitable for publication in IJMS with minor revision.

1.      Fig 1A; Fig 4A, B; Fig S2A should add the Bar.

2.      Some data lack statistical analysis. Such as, Fig 1B, Fig 6

3.      Which statistical method was used and what dose the different letter mean, which should be described in figure legends.

4.      I do not understand what is WRKY gene 30? WRKY gene 62? Jasmonate inducible pathogenesis-related class 10? In addition, the genes’ names in Fig6 legend is not accord with those in text. The prescriptive names of these genes should be used.

Author Response

  1. Fig 1A; Fig 4A, B; Fig S2A should add the Bar.

Response: Thanks for comments. Bars were added as required.

  1. Some data lack statistical analysis. Such as, Fig 1B, Fig 6

Response: Thanks for comments. The statistical analysis were performed in Fig 6, however, Fig 1B presented the expansion of necrosis lesions which are hard to mark with statistics in the figure.

  1. Which statistical method was used and what dose the different letter mean, which should be described in figure legends.

Response: After significant difference analysis, the difference results need to be marked. There are usually two ways, that is, marked with stars ( *) or letters (abcd) label. Both methods are both models can meet the needs of statistic. Stars may be more intuitive when there are fewer groups, however, letters are more concise when there are more groups. In this manuscript, ranking all the averages from the largest to the smallest, and then mark the letter a on the largest average. Then, compare the average with the following average numbers. If the difference is not significant, marking with the letter a, and the significant difference is marked with the letter b once a certain average number with. The average is marked with b as the standard, and comparing with the average above which is larger than it. All inconspicuous items are also marked with the letter b. If it is not significant, continue to mark it with the letter b until it encounters a significant difference with the average marked with c. The figure legends were detailed accordingly.

  1. I do not understand what is WRKY gene 30? WRKY gene 62? Jasmonate inducible pathogenesis-related class 10? In addition, the genes’ names in Fig6 legend is not accord with those in text. Theprescriptive names of these genes should be used.

Response: Thanks for comments. We apologize for our careless. All genes were updated in detail.

Reviewer 2 Report (New Reviewer)

The manuscript  Suppression of rice Osa-miR444.2 improves the resistance to  sheath blight in rice mediating through phytohormone pathway.

The authors reported that Osa-miR444b.2 negatively mediated the resistance to R. solani in rice, which will contribute to the cultivation of sheath blight re sistant varieties

I have no comments regarding the manuscript itself, or its organization. In my opinion, the article in the presented version is suitable for publication after minor revisions.

General comments

Is it possible to talk more comprehensively about sheath blight and it's economic losses on rice yield ( introduction section).

Author Response

Response: We appreciate this valuable comment. We added related information in the introduction section.

Reviewer 3 Report (New Reviewer)

Dear Colleagues

There are several questions about the used transgenic lines. Besides, it would be interesting to know your assumptions regarding the molecular mechanism of miRNA action.

Could you explain why you chose these lines In the case of the susceptible variety Xu3 - these are OE 3 and OE 4, in the case of the resistant variety YSBR1 OE 2 and OE 5?

Author Response

There are several questions about the used transgenic lines. Besides, it would be interesting to know your assumptions regarding the molecular mechanism of miRNA action.

Response: This project was initiated in our previous study (Cao, W.L.; Cao, X.X.; Zhao, J.H.; Zhang, Z.Y.; Feng, Z.M.; Ouyang, S.Q.; Zuo, S.M. Comprehensive characteristics of micro-RNA expression profile conferring to Rhizoctonia solani in rice. Rice Science. 2020, 27,101-112). By transcriptome analysis in this study, we found Osa-miR444b.2 mediated the response to R. solani infection by affecting both plant hormone signal transduction pathways and transcription factors. Plant hormones have been extensively studied for their importance in innate immunity particularly in the dicotyledonous model plant Arabidopsis thaliana. However, only in the last decade, plant hormones were demonstrated to play conserved and divergent roles in fine-tuning immune in rice. Transcription factor (TF) such as WRKY belongs to one of the major plant protein super-families. The WRKY TF gene family plays an important role in the regulation of transcriptional reprogramming associated with plant stress responses. TF play a key role in many types of environmental signals, including drought, temperature, salinity, cold, and biotic stresses. Therefore, we propose potential mechanisms behind Osa-miR444b.2 regulating the expression of critical resistant gene could be a useful strategy to improve disease resistance of crops.

Could you explain why you chose these lines In the case of the susceptible variety Xu3 - these are OE 3 and OE 4, in the case of the resistant variety YSBR1 OE 2 and OE 5?

Response: To investigate the biological function of Osa-miR444b.2 in rice immunity, we constructed the transgenic lines over-expressing Osa-miR444b.2 in Xu3 and YSBR1 genetic background, respectively. Actually, YSBR1 is a rice cultivar extraordinary resistance to R. solani. We designed both transgenic lines to compare the impact of over-expressing Osa-miR444b.2 in different genetic background. By qRT-PCR, the top transcriptional level two lines, OE 3 and OE 4 in Xu3, OE 2 and OE 5 in YSBR1, were selected for further study.  

Reviewer 4 Report (New Reviewer)

Dear authors

In this manuscript, you identified Osa-miR444.2 as a negative regulator and justified it through a number of experiments such as over-expressing this gene in the resistant cultivar and knocking it out in the susceptible cultivar. It is relevant and interesting to the readers to further manipulate this gene for sheath blight resistance in rice. The text is clear and easy to understand by a layman.

The research is original and it added a new regulator in the pathway of the sheath blight resistance mechanism although other negative regulators are known from previous studies such as Osmate 6 and BADH2 gene.

The conclusions drawn are consistent with the evidences presented and you successfully addressed the role of Osa-miR444.2 in sheath blight resistance.

The paper has moderate novelty and the results are consistent with number of experiments so recommended for publication.

Go through the comments enclosed for amendments in the revised manuscript.

Author Response

Go through the comments enclosed for amendments in the revised manuscript.

Response: We appreciate this valuable comment. We have revised the highlighted text point-to-point.

This manuscript is a resubmission of an earlier submission. The following is a list of the peer review reports and author responses from that submission.

Round 1

Reviewer 1 Report

Dear authors

Good work, congrats. I had few comments/suggestions which are given below.

Suggestions

The title “Suppression of rice Osa-miR444.2 improves the resistance to sheath blight in rice by mediating phytohormone pathway” can be changed to “Suppression of rice Osa-miR444.2 improves the resistance to sheath blight in rice mediating through phytohormone pathway”

Line 37

“second immunity” can be changed to “secondary immunity”

Line 50

“strategies to preventing” can be changed to “strategy to prevent”

Line 51

“MicroRNAs (miRNAs) are a class of 21-24 nucleotide single stranded non-coding” can be changed to “MicroRNAs (miRNAs) are a class non-coding single stranded RNAs usually with 21-24 nucleotides”

Line 78

“strategy” seems inappropriate usage it can be changed to “target”

Line 101

“induced upon to the R. solani infection” can be changed to “induced upon R. solani infection”

Line 103

“knockouting” change to “knocking out”

Line 137

grain weight or 1000- grain weight?

Line 146

“dependently” to “dependent”

Line 157

Knockouting to Knockouts

Comments

References are missing in  the text. Please recheck and verify the references in the text.

Osa-miR444b.2 was involved in the development of rice agronomic traits but is likely cultivar dependent. Too early to generalize without testing other cultivars representing the genetic diversity in rice.

It seems that Osa-miR444b.2 negatively affect the yield components then how yield will be improved. Does author record the yield per plant and measured the quality of the rice? If so, it can be included in the supplementary to substantiate line 26, 27 and 28.

The manuscript needs English editing.

Thank you

Author Response

Comments and Suggestions for Authors

Dear authors

Good work, congrats. I had few comments/suggestions which are given below.

Suggestions

The title “Suppression of rice Osa-miR444.2 improves the resistance to sheath blight in rice by mediating phytohormone pathway” can be changed to “Suppression of rice Osa-miR444.2 improves the resistance to sheath blight in rice mediating through phytohormone pathway”

Line 37

“second immunity” can be changed to “secondary immunity”

Thanks for this comment. We have done with the change accordingly.

Line 50

“strategies to preventing” can be changed to “strategy to prevent”

Thanks for this comment. We have done with the change accordingly.

Line 51

“MicroRNAs (miRNAs) are a class of 21-24 nucleotide single stranded non-coding” can be changed to “MicroRNAs (miRNAs) are a class non-coding single stranded RNAs usually with 21-24 nucleotides”

Thanks for this comment. We have done with the change accordingly.

Line 78

“strategy” seems inappropriate usage it can be changed to “target”

Thanks for this comment. We have done with the change accordingly.

Line 101

“induced upon to the R. solani infection” can be changed to “induced upon R. solani infection”

Thanks for this comment. We have done with the change accordingly.

Line 103

“knockouting” change to “knocking out”

Thanks for this comment. We have done with the change accordingly in the entire text.

Line 137

grain weight or 1000- grain weight?

Thanks for this comment. We changed “grain weight” with “1000-grain weight” accordingly.

Line 146

“dependently” to “dependent”

Thanks for this comment. We have done with the change accordingly.

Line 157

Knockouting to Knockouts

Thanks for this comment. We have done with the change accordingly.

Comments

References are missing in the text. Please recheck and verify the references in the text.

We apologize for our careless. All references were amended accordingly.

Osa-miR444b.2 was involved in the development of rice agronomic traits but is likely cultivar dependent. Too early to generalize without testing other cultivars representing the genetic diversity in rice.

We agree with this critical judgment. We have made a speculative statement of this conclusion based on the data currently available.

It seems that Osa-miR444b.2 negatively affect the yield components then how yield will be improved. Does author record the yield per plant and measured the quality of the rice? If so, it can be included in the supplementary to substantiate line 26, 27 and 28.

We appreciate this comment. Actually, we had according field experiment once, but we regret to say that we could not obtain the expected production statistics due to the unpredictable natural conditions including the current COVID-19 pandemic. We have future plans to investigate how Osa-miR444b.2 negatively affects the yield components in rice.

The manuscript needs English editing.

Thanks for this comment. We have invited an English native language expert to revise this manuscript entirely.

Reviewer 2 Report

The paper of Feng et al. discussed the role of Osa-miR444.2 in sheath blight resistance. The paper might be interesting to the readers. However, the current organization of the paper is not acceptable for quality publication. For one, the references in the paper are a total mess. A lot of "Error! Reference source not found" can be seen in the paper. Similarly, there are unnecessary numbers in the paper. For example, in line 231 "plant host 32", and in line 234 "plant 17". Therefore, my recommendation for this paper is rejection. 

Author Response

Comments and Suggestions for Authors

The paper of Feng et al. discussed the role of Osa-miR444.2 in sheath blight resistance. The paper might be interesting to the readers. However, the current organization of the paper is not acceptable for quality publication. For one, the references in the paper are a total mess. A lot of "Error! Reference source not found" can be seen in the paper. Similarly, there are unnecessary numbers in the paper. For example, in line 231 "plant host 32", and in line 234 "plant 17". Therefore, my recommendation for this paper is rejection.

We appreciate these comments, and we DO apologize for our careless. We have invited an English native language expert to revise this manuscript entirely.